# Towards analyzing self-attention via linear neural networks

## Abstract

Self-attention is a key component of the transformer architecture which has driven much of recent advances in AI. Theoretical analysis of self-attention has received significant attention and remains a work in progress. In this paper, we analyze gradient flow training of a simplified transformer model consisting of a single linear self-attention layer (thus it lacks softmax, MLP, layer-normalization, and positional information) with a single head on a histogram-like problem: the input is a sequence of symbols from an alphabet and the output is a sequence of the same length with each position consisting of the frequency in the input sequence of the symbol at that position in the input sequence. Our analysis goes via a reduction to 2-layer linear neural networks in which the input layer matrix is a diagonal matrix. We provide a complete analysis of gradient flow on these networks. Our reduction to linear neural networks involves one assumption which we empirically verify. Our analysis applies to various extensions of the histogram problem.

## 1 Introduction

Self-attention is a key component in transformers Vaswani et al. (2017) which have radically transformed NLP (OpenAI, 2023), vision (Dosovitskiy et al., 2021), multimodal (Jaegle et al., 2021) and more. Unfortunately, theoretical understanding of transformers, and in particular of self-attention, lags far behind these dramatic advances.

Many recent works study transformers and self-attention theoretically from various viewpoints including their expressive and computational power. Here we confine our discussion to papers on training dynamics. Jelassi et al. (2022); Li et al. (2023); Tian et al. (2023); Boix-Adserà et al. (2023); Zhang et al. (2023); Tarzanagh et al. (2023) analyze training dynamics of transformers for various specific problems. While insightful, these papers generally involve stylized assumptions and this makes it difficult to compare the results.

Our work is in this line of research but differs from the prior work in the specific problems, assumptions, and proof method. We analyze gradient flow training of a simplified transformer model consisting of a single linear self-attention layer (thus it lacks softmax, MLP, layer-normalization, and positional information) with a single head on the histogram problem (Weiss et al., 2021): the input is a sequence of symbols from an alphabet and the output is a sequence of the same length with each position consisting of the frequency in the input of the symbol at that position in the input sequence. For example, for the input sequence $adbcba$, the output sequence would be $[2, 1, 2, 1, 2, 2]$. In addition to the basic histogram problem, we also consider its extensions. These extensions, instead of counting the number of times a symbol occurs in the input, count the number of symbols in the input that stand in a specific relation to the character. These problems require nontrivial learning: the model needs to learn relations among the symbols and count the relevant symbols (for basic histograms this relation is the equality relation but we allow more complex relations). These histogram-like problems can be solved by the simplified transformer as we show via constructions.

Our method is via reduction of training of simplified transformer on the histogram problem to the training of a linear neural network with two layers. Linear neural networks are neural networks without nonlinearities; in other words, the output $y$ is obtained by successively applying linear transformations to the input: $y = W_1 \dots W_L x$. Clearly, linear neural networks are highly limited as they are only able to compute linear functions of the input. However, their loss landscape is non-convex in the parameters though they tend to be relatively more tractable than nonlinear neural

networks. Moreover, their training dynamics can exhibit the richness of (nonlinear) neural networks (Saxe et al., 2019; Kleinman et al., 2023). These characteristics have led to intense study of linear neural networks, e.g., (Baldi & Hornik, 1989; Saxe et al., 2014; Kawaguchi, 2016; Arora et al., 2019; Du & Hu, 2019; Zou et al., 2020; Min et al., 2021; Xu et al., 2023; Min et al., 2023). While this work has greatly increased the understanding of linear neural networks, to our knowledge, analysis of training of 2-layer linear networks remains open without significant assumptions.

The linear neural networks arising in our work have two layers and have a special structure: $\boldsymbol{y} = \boldsymbol{W_1}\boldsymbol{W_2}\boldsymbol{x}$, with $\boldsymbol{W_1}$ and $\boldsymbol{W_2}$ square matrices and $\boldsymbol{W_2}$ a diagonal matrix. We call these networks *df-linear networks* for diagonal layer followed by the fully-connected layer. While df-linear networks seem to not have been considered in the literature before, simpler diagonal linear networks where both layers are diagonal have been. Even these networks are sufficiently rich to exhibit many features of the training dynamics of nonlinear networks and have seen significant work recently, e.g., Vaskevicius et al. (2019); Woodworth et al. (2020); HaoChen et al. (2021); Pesme et al. (2021); Berthier (2023); Boix-Adserà et al. (2023). The corresponding optimization problem remains non-convex and challenging, though the added simplicity of both layers being diagonal now allows for studying many finer aspects of training such as the effect of noise and incremental learning (how the complexity of the learned networks changes during training).

While the requirement of square matrices in df-linear network precludes use of large width to make analysis tractable, $\boldsymbol{W_2}$ being diagonal allows for decoupling of different coordinates reducing the problem to a collection of 1-dimensional problems. We give an essentially complete analysis of the 1-dimensional problem.

## 2 PRELIMINARIES

### 2.1 NOTATIONS

The set $\{1, 2, \ldots, d\}$ is denoted by $[d]$. If $f$ is a time-dependent function, then $f_t$ and $f(t)$ are interchangeably used to denote the value of $f$ at time $t$. The time derivative $\frac{\mathrm{d}f}{\mathrm{d}t}$ is also denoted by $\dot{f}$.

We define the map Diag $: \mathbb{R}^d \to \mathbb{R}^{d \times d}$ that takes a vector $\boldsymbol{a} = [a_1, \ldots, a_d]^T$ and produces a diagonal matrix Diag($\boldsymbol{a}$) whose $i$-th diagonal entry is $a_i$. Similarly, the map diag $: \mathbb{R}^{d \times d} \to \mathbb{R}^d$ takes a matrix $\boldsymbol{A} = [A_{i,j}]$ as an input and produces a $d$-dimensional vector diag($\boldsymbol{A}$) whose $i$-th coordinate is $A_{ii}$. We also define the function $\mathcal{D} : \mathbb{R}^{d \times d} \to \mathbb{R}^{d \times d}$ by $\mathcal{D}(\boldsymbol{A}) = \text{Diag}(\text{diag}(\boldsymbol{A}))$; i.e., it retrieves the diagonal part of the matrix. We use $\mathbb{1}$ to denote an indicator; for an expression $e$, $\mathbb{1}_e$ is 1 if $e$ is true, and 0 if $e$ is false.

For a vector $v$, $\|v\|$ denotes the $L_2$ norm. For a matrix $A$, $\|A\|_F$ denotes the Frobenius norm.

### 2.2 TRANSFORMERS

We consider a variant of the transformer architecture given in Vaswani et al. (2017). Let the input alphabet be $\Sigma$. We model sequence-to-sequence functions of the form $f : \Sigma^N \to \mathbb{R}^{N \times d_{\text{out}}}$, using attention-only 1-layer transformers.

The input sequence is given to the transformer as a matrix of one-hot embeddings of the sequence, $\boldsymbol{X} \in \mathbb{R}^{N \times |\Sigma|}$, where the $i$-th row of $\boldsymbol{X}$ corresponds to the one-hot embedding of the $i$-th element in the sequence. A single layer in a transformer consists of the following parameters(where $d_{\text{model}}, d_{\text{out}}$ are hyperparameters):

1 Embedding Matrix: $\boldsymbol{W}_E \in \mathbb{R}^{|\Sigma| \times d_{\text{emb}}}$

2 Query Weights: $\boldsymbol{W}_Q \in \mathbb{R}^{d_{\text{emb}} \times d_{\text{model}}}$

3 Key Weights: $\boldsymbol{W}_K \in \mathbb{R}^{d_{\text{emb}} \times d_{\text{model}}}$

4 Value Weights: $\boldsymbol{W}_V \in \mathbb{R}^{d_{\text{emb}} \times d_{\text{model}}}$

5 Output Weights: $\boldsymbol{W}_F \in \mathbb{R}^{d_{\text{model}} \times d_{\text{out}}}$.

Let $\boldsymbol{E} = \boldsymbol{X}\boldsymbol{W}_E$. Then forward pass through this layer is given by: $\boldsymbol{Y} = \boldsymbol{E}\boldsymbol{W}_Q\boldsymbol{W}_K^T\boldsymbol{E}^T\boldsymbol{E}\boldsymbol{W}_V\boldsymbol{W}_F$.

To make the analysis tractable, we consider further simplifications of the above model. Similar simplifications are made in many other theoretical studies. First, we assume $d_{\text{emb}} = |\Sigma| = d$, and fix $\boldsymbol{W}_E$ to be $\boldsymbol{I}_d$, the identity matrix. Further, instead of considering four separate learnable parameters $(\boldsymbol{W}_Q, \boldsymbol{W}_K, \boldsymbol{W}_V, \boldsymbol{W}_F)$, we consider only two, $\boldsymbol{W}_{QK} = \boldsymbol{W}_Q \boldsymbol{W}_K^T \in \mathbb{R}^{d \times d}$ and $\boldsymbol{W}_{VF} = \boldsymbol{W}_V \boldsymbol{W}_F \in \mathbb{R}^{d \times d_{\text{out}}}$, since the output of the transformer depends only on the products of the matrices. Thus, the output of our model is given by:

$$\boldsymbol{Y} = \boldsymbol{X} \boldsymbol{W}_{QK} \boldsymbol{X}^T \boldsymbol{X} \boldsymbol{W}_{VF}.$$

From here on, we write $\boldsymbol{Q} = \boldsymbol{W}_{QK}$ and $\boldsymbol{V} = \boldsymbol{W}_{VF}$.

Given a set of input-output pairs $\mathcal{D}$ for the task, training consists of minimizing a loss function $\ell : \mathbb{R}^{d \times d} \times \mathbb{R}^{d \times d_{\text{out}}} \to \mathbb{R}$. Here, we consider the squared-norm loss, given by

$$\ell(\boldsymbol{Q}, \boldsymbol{V}) = \frac{1}{2|\mathcal{D}|} \sum_{(\boldsymbol{X}, \boldsymbol{Y}) \in \mathcal{D}} \left\| \boldsymbol{X} \boldsymbol{Q} \boldsymbol{X}^T \boldsymbol{X} \boldsymbol{V} - \boldsymbol{Y} \right\|_F^2.$$

## 3 FROM SELF-ATTENTION TO LINEAR NEURAL NETWORKS FOR HISTOGRAMS

In this section we introduce the histogram tasks, and then present our reduction from self-attention to linear neural networks.

### 3.1 HISTOGRAM TASKS

In general terms, we define histogram-like tasks as those where, given an input sequence of symbols from an alphabet $\Sigma$, the output sequence is determined based on the frequency of each symbol's appearance within the input, rather than their specific order or arrangement. These tasks are well-suited for transformers, allowing us to outline specific constructions of transformers that effectively solve these tasks. Below, we discuss several examples of such tasks, followed by their transformer constructions.

1. `Equality Histogram`: For each symbol $s$ in the sequence, the total number of symbols in the sequence with same value as $s$ is computed, which forms the output sequence. For example, for the input sequence $adbcba$, the output sequence would be $[2, 1, 2, 1, 2, 2]$. Concretely, let $\Sigma = \{a_1, \ldots, a_d\}$. Given an $N$-length sequence $s_1 s_2 \ldots s_N \in \Sigma^N$, we want to output $[c_{s_1}, c_{s_2}, \ldots, c_{s_N}]$, where $c_x = \sum_{i \in [N]} \mathbb{1}_{x=s_i}$, i.e., the count of the number of times $x$ appears in the string. This problem was considered in Weiss et al. (2021).

2. `Less than Histogram`: We define an ordering on the alphabet. For each symbol $s$ in the sequence, the total number of symbols in the sequence which have value less than or equal to $s$ according to the ordering are computed, and form the output sequence. For example, given the input sequence $adbcba$, and the standard lexicographical ordering, the output sequence will be $[2, 6, 4, 5, 4, 2]$.
   Formally, we define an ordering $\preceq$ on $\Sigma$, and given an input string $s_1 s_2 \ldots s_N$, the output is the sequence $[l_{s_1}, l_{s_2}, \ldots, l_{s_N}]$, where $l_x = \sum_{i \in [N]} \mathbb{1}_{s_i \preceq x}$.

3. `Threshold Histogram`: For this task, we define an ordering on the alphabet and a threshold $D \in \mathbb{N}$. For each symbol $s$ in the sequence, we compute the number of symbols that are at a distance of at most $D$ according to the ordering, and form the output sequence. For example, for $D = 2$ and the standard lexicographical ordering, for input $adbcba$, the output will be $[5, 4, 6, 6, 6, 5]$.

### 3.2 TRANSFORMER CONSTRUCTIONS FOR HISTOGRAMS

The input sequence will be given to the transformer in the form of a matrix of one-hot embeddings. Formally, given an alphabet $\Sigma = \{a_1, a_2, \ldots, a_d\}$, and the input sequence $s_1, s_2, \ldots, s_N$, the transformer input is a matrix $\boldsymbol{X} \in \mathbb{R}^{N \times d}$, where the $i$-th row of $\boldsymbol{X}$ is $\boldsymbol{e}_{s_i} \in \mathbb{R}^d$.

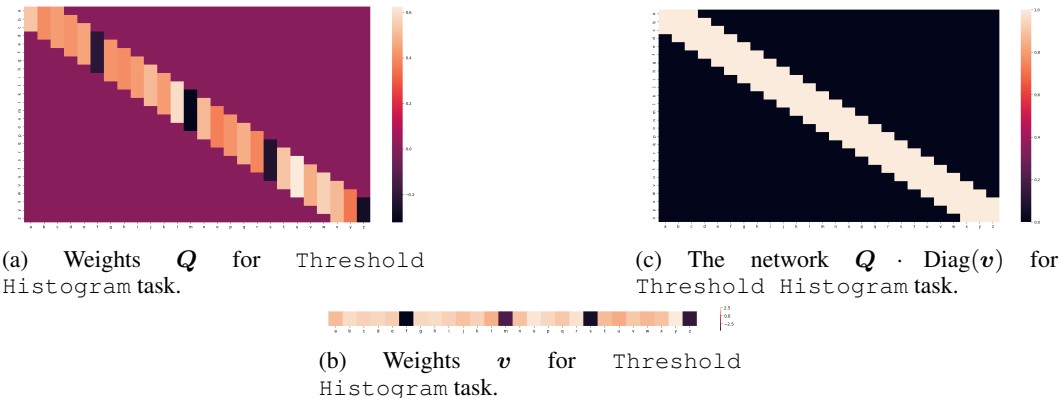

(a) Weights $\boldsymbol{Q}$ for `Threshold Histogram` task.

(c) The network $\boldsymbol{Q} \cdot \text{Diag}(\boldsymbol{v})$ for `Threshold Histogram` task.

(b) Weights $\boldsymbol{v}$ for `Threshold Histogram` task.

Figure 1: Learned weights for `Threshold Histogram` task with threshold $D = 2$, where the input alphabet is the set of lowercase English letters with the lexicographical ordering. The learned network $\boldsymbol{Q}\text{Diag}(\boldsymbol{v})$ matches $\boldsymbol{M}_{\text{thr}}$, as predicted by our analysis.

For all of the tasks, $d_{\text{out}} = 1$, so the parameters are just $\boldsymbol{Q} \in \mathbb{R}^{d \times d}$ and $\boldsymbol{v} \in \mathbb{R}^d$. The constructions we present are not unique: given a solution $(\boldsymbol{Q}, \boldsymbol{v})$, we can always scale $\boldsymbol{Q}$ by some non-zero $\alpha \in \mathbb{R}$ and $\boldsymbol{v}$ by $1/\alpha$, to get a whole family of solutions. In fact, we can scale each column of $\boldsymbol{Q}$ by a different non-zero $\alpha \in \mathbb{R}$, and scale the corresponding entry in $\boldsymbol{v}$ by $1/\alpha$ to get a new solution. We will show later that all solutions of histogram-like tasks admit this form, and hence the solutions are unique upto this symmetry.

Below we give the constructions for each of the above described tasks.

1. `Equality Histogram`: Consider $\boldsymbol{Q}_{\text{eq}} = \boldsymbol{I}$, the identity matrix, and $\boldsymbol{v}_{\text{eq}} = \boldsymbol{1}_d$, the vector of all ones. Then, if $\boldsymbol{y} = \boldsymbol{X}\boldsymbol{Q}_{\text{eq}}\boldsymbol{X}^T\boldsymbol{X}\boldsymbol{v}_{\text{eq}} = \boldsymbol{X}\boldsymbol{X}^T\boldsymbol{X}\boldsymbol{1}_d$, we have

$$y_i = (\boldsymbol{X}\boldsymbol{X}^T\boldsymbol{1}_N)_i = \sum_{j=1}^N \boldsymbol{e}_{s_i}^T \boldsymbol{e}_{s_j} = \sum_{j=1}^N \mathbb{1}_{s_i = s_j}.$$

2. `Less than Histogram`: Assume that the ordering is such that $a_i \preceq a_j \iff i \leq j$ (for other orderings, we can just permute the constructions given here). Then, $\boldsymbol{v}_{\text{lt}} = \boldsymbol{1}_d$, and $\boldsymbol{Q}_{\text{lt}} = \boldsymbol{L}_d$, where $\boldsymbol{L}_d$ is a lower triangular matrix of all ones i.e. $\boldsymbol{L}_d(i, j) = 1$ if $i \geq j$ else it is 0. Thus, if $\boldsymbol{y} = \boldsymbol{X}\boldsymbol{Q}_{\text{lt}}\boldsymbol{X}^T\boldsymbol{X}\boldsymbol{v}_{\text{lt}}$, we have

$$y_i = (\boldsymbol{X}\boldsymbol{L}_d\boldsymbol{X}^T\boldsymbol{1}_N)_i = \sum_{j=1}^N \boldsymbol{e}_{s_i}^T \boldsymbol{L}_d \boldsymbol{e}_{s_j} = \sum_{j=1}^N \boldsymbol{L}_d(s_i, s_j) = \sum_{j=1}^N \mathbb{1}_{s_j \preceq s_i}.$$

3. `Threshold Histogram`: Define $\boldsymbol{v}_{\text{thr}} = \boldsymbol{1}_d$ and $\boldsymbol{Q}_{\text{thr}} = \boldsymbol{T}_d$, where $\boldsymbol{T}_d \in \mathbb{R}^{d \times d}$ is such that $\boldsymbol{T}_d(i, j) = 1$ if $|i - j| \leq D$, otherwise $\boldsymbol{T}_d(i, j) = 0$. It can be verified that this construction solves the `Threshold Histogram` task.

## 3.3 REDUCTION TO DF-LINEAR NETWORKS

A unifying feature of each of the above described tasks is that the output depends only on the frequency of each of the symbols from the alphabet present, and does not depend on the order in which they appear. Therefore, the output can be described by a function $g : \mathbb{R}^d \to \mathbb{R}^N$, where the input to $g$ is $\boldsymbol{c}$, where $\boldsymbol{c}$ is the vector of frequencies of symbols in the input i.e. $c_i$ is the number of times the $i$-th symbol appears in the input. In fact, all the above tasks also depend *linearly* on $\boldsymbol{c}$, and we can write $g(\boldsymbol{c}) = \boldsymbol{X}\boldsymbol{M}\boldsymbol{c}$, for some matrix $\boldsymbol{M} \in \mathbb{R}^{d \times d}$. This can also be seen from the constructions above. For example, for the `Equality Histogram` task, the output can be written in terms of the frequency vector as $g_{\text{eq}}(\boldsymbol{c}) = \boldsymbol{X}\boldsymbol{c}$, and so, $\boldsymbol{M}_{\text{eq}} = \boldsymbol{I}$. For the `Less than Histogram` task, the output can be written as $g_{\text{lt}}(\boldsymbol{c}) = \boldsymbol{X}\boldsymbol{L}_d\boldsymbol{c}$, and $\boldsymbol{M}_{\text{lt}} = \boldsymbol{L}_d$, and similarly, for the `Threshold Histogram` task, we have $\boldsymbol{M}_{\text{thr}} = \boldsymbol{T}_d$.

The above fact allows us to reduce the training dynamics of transformers trained on histogram-like tasks, to the dynamics of relatively simpler networks which we call *DF-Linear Networks*. These networks are linear networks where the first layer is constrained to be a diagonal matrix, and the next layer is a fully connected layer.

**Lemma 3.1.** *Consider a histogram task with the objective function given by $g(\boldsymbol{c}) = \boldsymbol{X}\boldsymbol{M}\boldsymbol{c}$ for a matrix $\boldsymbol{M} \in \mathbb{R}^{d \times d}$, where $\boldsymbol{c} \in \mathbb{R}^d$ is the vector of frequencies. Then, for a dataset of input-output pairs $\mathcal{D}$, we have*

$$\ell_{\mathrm{tf}}(\boldsymbol{Q}, \boldsymbol{v}) = \frac{1}{2|\mathcal{D}|} \sum_{(\boldsymbol{X},\boldsymbol{y}) \in \mathcal{D}} \left\| \boldsymbol{X}\boldsymbol{Q}\boldsymbol{X}^T\boldsymbol{X}\boldsymbol{v} - \boldsymbol{y} \right\|^2$$

$$= \frac{1}{2|\mathcal{D}|} \sum_{(\boldsymbol{X},\boldsymbol{y}) \in \mathcal{D}} \left\| \mathrm{Diag}(\boldsymbol{c}_{\boldsymbol{X}})^{1/2} \left( \boldsymbol{Q}\mathrm{Diag}(\boldsymbol{v}) - \boldsymbol{M} \right) \boldsymbol{c}_{\boldsymbol{X}} \right\|^2,$$

*where $\boldsymbol{c}_{\boldsymbol{X}}$ is the vector of frequencies associated to the input $\boldsymbol{X}$.*

*Proof.* Observe that for input $(\boldsymbol{X}, \boldsymbol{y}) \in \mathcal{D}$ we have $\boldsymbol{X}^T\boldsymbol{X} = \mathrm{Diag}(\boldsymbol{c}_{\boldsymbol{X}})$, and $\boldsymbol{y} = \boldsymbol{X}\boldsymbol{M}\boldsymbol{c}_{\boldsymbol{X}}$. Thus, we can write the transformer loss as

$$\ell_{\mathrm{tf}}(\boldsymbol{Q}, \boldsymbol{v}) = \frac{1}{2|\mathcal{D}|} \sum_{(\boldsymbol{X},\boldsymbol{y}) \in \mathcal{D}} \left\| \boldsymbol{X}\boldsymbol{Q}\boldsymbol{X}^T\boldsymbol{X}\boldsymbol{v} - \boldsymbol{y} \right\|^2$$

$$= \frac{1}{2|\mathcal{D}|} \sum_{(\boldsymbol{X},\boldsymbol{y}) \in \mathcal{D}} \left\| \boldsymbol{X}\boldsymbol{Q}\mathrm{Diag}(\boldsymbol{c}_{\boldsymbol{X}})\boldsymbol{v} - \boldsymbol{X}\boldsymbol{M}\boldsymbol{c}_{\boldsymbol{X}} \right\|^2$$

$$= \frac{1}{2|\mathcal{D}|} \sum_{(\boldsymbol{X},\boldsymbol{y}) \in \mathcal{D}} \left\| \boldsymbol{X}\boldsymbol{Q}\mathrm{Diag}(\boldsymbol{v})\boldsymbol{c}_{\boldsymbol{X}} - \boldsymbol{X}\boldsymbol{M}\boldsymbol{c}_{\boldsymbol{X}} \right\|^2$$

$$= \frac{1}{2|\mathcal{D}|} \sum_{(\boldsymbol{X},\boldsymbol{y}) \in \mathcal{D}} \left\| \boldsymbol{X}(\boldsymbol{Q}\mathrm{Diag}(\boldsymbol{v}) - \boldsymbol{M})\boldsymbol{c}_{\boldsymbol{X}} \right\|^2$$

$$= \frac{1}{2|\mathcal{D}|} \sum_{(\boldsymbol{X},\boldsymbol{y}) \in \mathcal{D}} \boldsymbol{c}_{\boldsymbol{X}}^T(\boldsymbol{Q}\mathrm{Diag}(\boldsymbol{v}) - \boldsymbol{M})^T\mathrm{Diag}(\boldsymbol{c}_{\boldsymbol{X}})(\boldsymbol{Q}\mathrm{Diag}(\boldsymbol{v}) - \boldsymbol{M})\boldsymbol{c}_{\boldsymbol{X}})$$

$$= \frac{1}{2|\mathcal{D}|} \sum_{(\boldsymbol{X},\boldsymbol{y}) \in \mathcal{D}} \left\| \mathrm{Diag}(\boldsymbol{c}_{\boldsymbol{X}})^{1/2} \left( \boldsymbol{Q}\mathrm{Diag}(\boldsymbol{v}) - \boldsymbol{M} \right) \boldsymbol{c}_{\boldsymbol{X}} \right\|^2.$$

$\square$

The above lemma allows us to characterize the solution sets of the histogram tasks exactly. The loss function will be zero only when $\mathrm{Diag}(\boldsymbol{c}_{\boldsymbol{X}})^{1/2} \left( \boldsymbol{Q}\mathrm{Diag}(\boldsymbol{v}) - \boldsymbol{M} \right) \boldsymbol{c}_{\boldsymbol{X}} = 0$ for all $(\boldsymbol{X}, \boldsymbol{y}) \in \mathcal{D}$. For a randomly sampled dataset, this would imply that, almost surely, $\boldsymbol{Q}\mathrm{Diag}(\boldsymbol{v}) = \boldsymbol{M}$ or equivalently, $v_i\boldsymbol{q}_i = \boldsymbol{m}_i$ for all $i \in [d]$, where $\boldsymbol{q}_i$ (resp. $\boldsymbol{m}_i$) is the $i$-th column of $\boldsymbol{Q}$ (resp. $\boldsymbol{M}$). Figure 1 show visualizations of learned parameters for a transformer trained on the `Threshold Histogram` task. Note that the learned $\boldsymbol{Q}$ and $\boldsymbol{v}$ parameters belong to the above described solution set.

In Section 4, we analyze the training dynamics of DF-Linear Networks, i.e. gradient flow on the objective $\ell_{\mathrm{df}}(\boldsymbol{Q}, \boldsymbol{v}) = \frac{1}{2} \left\| \boldsymbol{Q} \cdot \mathrm{Diag}(\boldsymbol{v}) - \boldsymbol{M} \right\|_F^2$. This is a simplification from the full histogram objective as derived in Lemma 3.1 and subject of the following assumption.

**Assumption 3.1.** *Let $\mathcal{D}$ be a randomly sampled dataset for a histogram task. Let $(\boldsymbol{Q}_{\mathrm{tf}}(t), \boldsymbol{v}_{\mathrm{tf}}(t))$ (resp. $(\boldsymbol{Q}_{\mathrm{df}}(t), \boldsymbol{v}_{\mathrm{df}}(t))$) be the parameters for the transformers at time $t$, under gradient flow on $\ell_{\mathrm{tf}}$ (resp. $\ell_{\mathrm{df}}$), starting from $(\boldsymbol{Q}_0, \boldsymbol{v}_0)$. Then, with high probability, there exist constants $C, \sigma, T > 0$, depending on the data $\mathcal{D}$, such that, for all $t \geq T$,*

$$\ell_{\mathrm{tf}}(\boldsymbol{Q}_{\mathrm{tf}}(t), \boldsymbol{v}_{\mathrm{tf}}(t)) \leq C\ell_{\mathrm{df}}(\boldsymbol{Q}_{\mathrm{df}}(\sigma t), \boldsymbol{v}_{\mathrm{df}}(\sigma t)).$$

Assumption 3.1 completes the reduction of the transformer dynamics to the DF-Linear Networks dynamics. Note that if the gradient flow on the transformer loss converges to a zero loss solution $(\boldsymbol{Q}_{\mathrm{tf}}^*, \boldsymbol{v}_{\mathrm{tf}}^*)$, then we would have $\boldsymbol{Q}_{\mathrm{tf}}^* \cdot \mathrm{Diag}(\boldsymbol{v}_{\mathrm{tf}}^*) = \boldsymbol{M}$. Similarly, if gradient flow on the DF-Linear

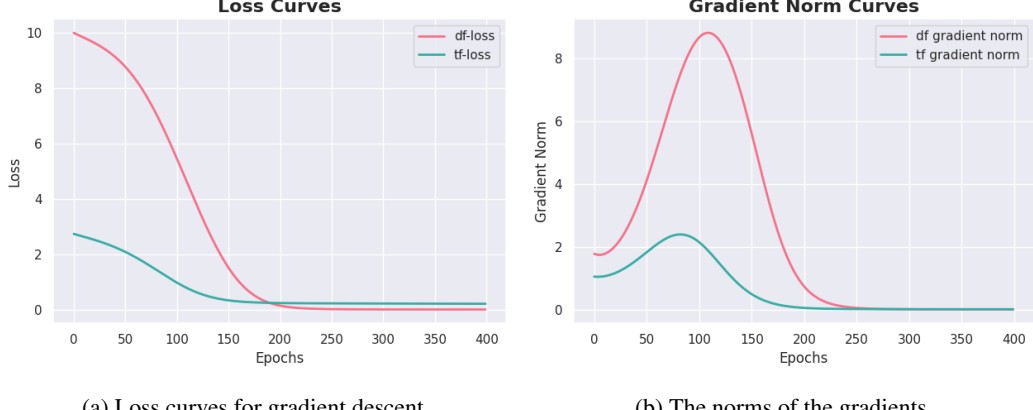

(a) Loss curves for gradient descent.

(b) The norms of the gradients.

Figure 2: Loss and the norm of the gradient for gradient descent on $\ell_{\text{tf}}$ and $\ell_{\text{df}}$, starting from the same initialization. The dynamics of $\ell_{\text{tf}}$ are just scaled dynamics of $\ell_{\text{df}}$.

Networks loss converges to a zero loss solution $(\boldsymbol{Q}^*_{\text{df}}, \boldsymbol{v}^*_{\text{df}})$, then $\boldsymbol{Q}^*_{\text{df}} \cdot \text{Diag}(\boldsymbol{v}^*_{\text{df}}) = \boldsymbol{M}$. In both the cases, due to the symmetries inherent in the objective function, we can derive a conserved quantity which further constrains the possible solutions. The conserved quantity, given by Lemma A.1, is the same for both loss functions, since the initializations are the same and both functions satisfy the same symmetries. The two constraints together have only a single solution; thus, we get that $(\boldsymbol{Q}^*_{\text{tf}}, \boldsymbol{v}^*_{\text{tf}}) = (\boldsymbol{Q}^*_{\text{df}}, \boldsymbol{v}^*_{\text{df}})$.

Assumption 3.1 holds true experimentally. We optimize $\ell_{\text{tf}}$ and $\ell_{\text{df}}$ using gradient descent (as an approximation of gradient flow), starting from the same initialization. For the transformer training, a dataset consisting of 100 random strings of length 100 were sampled, where each element of the string was uniformly sampled from the first 10 letters of the alphabet, and the task considered was the `Equality Histogram` task. Figure 2 depicts the training dynamics during optimization for the two loss functions. Note that, apart from a scaling factor, the dynamics are similar.

## 4 CONVERGENCE ANALYSIS OF DF-LINEAR NETWORKS

In this section we prove that gradient flow on the squared-norm loss for DF-Linear Networks converges, for all initializations, apart from a measure-zero set of initial points, which we characterize.

We consider the loss function $\ell_{\text{df}}(\boldsymbol{Q}, \boldsymbol{v}) = \frac{1}{2} \|\boldsymbol{Q}\text{Diag}(\boldsymbol{v}) - \boldsymbol{M}\|_F^2$, for a $d \times d$ target matrix $\boldsymbol{M}$. To ease the clutter of notations we refer to $\ell_{\text{df}}$ by just $\ell$.

The proof has two main steps. First, we show that the dynamics of the gradient flow can be decoupled across the dimensions of the parameters. This can be seen immediately if we write down the objective in the following form:

$$\ell(\boldsymbol{Q}, \boldsymbol{v}) = \frac{1}{2} \sum_{i \in [d]} \|v_i \boldsymbol{q}_i - \boldsymbol{m}_i\|^2 ,$$

where $\boldsymbol{q}_i$ is the $i$-th column of $\boldsymbol{Q}$. Following this, we reduce the dynamics along each dimension to a dynamical system, which can be analysed completely. Thus, we get the following theorem.

**Theorem 1.** *Consider the loss function* $\ell : \mathbb{R}^{d \times d} \times \mathbb{R}^d \to \mathbb{R}$ *defined as*

$$\ell(\boldsymbol{Q}, \boldsymbol{v}) = \frac{1}{2} \|\boldsymbol{Q} \cdot \text{Diag}(\boldsymbol{v}) - \boldsymbol{M}\|_F^2 .$$

*Consider the gradient flow on* $\ell$, *starting from* $(\boldsymbol{Q}_0, \boldsymbol{v}_0)$, *where* $\boldsymbol{Q}_0 \in \mathbb{R}^{d \times d}$ *and* $\boldsymbol{v}_0 \in \mathbb{R}^d$.

*Let* $\boldsymbol{Q}_0 = [\boldsymbol{q}_1^{(0)}, \ldots, \boldsymbol{q}_d^{(0)}]$, $\boldsymbol{M} = [\boldsymbol{m}_1, \ldots, \boldsymbol{m}_d]$ *and* $\boldsymbol{v}_0 = [v_1^{(0)}, \ldots, v_d^{(0)}]^T$, *and for each* $i \in [d]$ *let*

$$\delta_i = \frac{\left\|\boldsymbol{q}_i^{(0)}\right\|^2 - v_i^{(0)\,2}}{\|\boldsymbol{m}_i\|}, \quad \rho_i = \sqrt{1 + \frac{\delta_i^2}{4}} + \frac{\delta_i}{2}, \text{ and } \bar{\rho}_i = {}^1\!/\!_{\rho_i}.$$

*Further define the quantities $T, \alpha$ and $\gamma$ as*

$$T = \max_i \left\{ \ln \left( \bar{\rho}_i^2 \left| \frac{\rho_i v_i^{(0)} \|\boldsymbol{m}_i\| - \boldsymbol{q}_i^{(0)^T} \boldsymbol{m}_i}{\bar{\rho}_i v_i^{(0)} \|\boldsymbol{m}_i\| + \boldsymbol{q}_i^{(0)^T} \boldsymbol{m}_i} \right| \right) \right\} \text{ and}$$

$$\alpha = \min_i \{\|\boldsymbol{m}_i\| \, \bar{\rho}_i\} \text{ and } \gamma = \begin{cases} 1 & \text{if } \delta_i = 0 \text{ for some } i \\ 0 & \text{otherwise.} \end{cases}$$

*Then, provided that $-\bar{\rho}_i v_i^{(0)} \neq \boldsymbol{q}_i^{(0)^T} \boldsymbol{m}_i$ for all $i \in [d]$, the loss function $\ell$ exhibits the following decay for all $t > T$:*

$$\ell(t) = \mathcal{O}\left(d \max_i \|\boldsymbol{m}_i\|^2 \cdot t^\gamma \exp(-2\alpha t)\right).$$

*Proof.* The proof follows by reducing the gradient flow equations to a set of dynamical systems of the form described in Equation 2. Let $\boldsymbol{Q} = [\boldsymbol{q}_1 \cdots \boldsymbol{q}_d]$, $\boldsymbol{M} = [\boldsymbol{m}_1 \cdots \boldsymbol{m}_d]$ and $\boldsymbol{V} = \text{Diag}(\boldsymbol{v})$. Then, we have

$$\frac{\mathrm{d}\boldsymbol{Q}}{\mathrm{d}t} = -\frac{\partial \ell}{\partial \boldsymbol{Q}} = -(\boldsymbol{Q}\boldsymbol{V} - \boldsymbol{M})\boldsymbol{V},$$

$$\frac{\mathrm{d}\boldsymbol{v}}{\mathrm{d}t} = -\frac{\partial \ell}{\partial \boldsymbol{v}} = -\text{diag}(\boldsymbol{Q}^T(\boldsymbol{Q}\boldsymbol{V} - \boldsymbol{M})).$$

Abbreviating the time derivative $\frac{\mathrm{d}\boldsymbol{v}}{\mathrm{d}t}$ to $\dot{\boldsymbol{v}}$ etc., for each $i \in [d]$, we get

$$\dot{\boldsymbol{q}}_i = v_i(\boldsymbol{m}_i - v_i \boldsymbol{q}_i),$$

$$\dot{v}_i = \boldsymbol{q}_i^T \boldsymbol{m}_i - v_i \|\boldsymbol{q}_i\|^2.$$

Further, following lemma A.1 we note that $\mathcal{D}(\boldsymbol{Q}^T \boldsymbol{Q} - v_t v_t^T)$ is conserved throughout the trajectory. So, let the $i$-th diagonal entry of the conserved quantity at initialization be $\beta_i$. Then for all $t$ we have

$$\|\boldsymbol{q}_i\|^2 - v_i^2 = \beta_i, \tag{1}$$

for some constants $\beta_i \in \mathbb{R}$. Using this, we get the system of equations:

$$\dot{\boldsymbol{q}}_i = v_i(\boldsymbol{m}_i - v_i \boldsymbol{q}_i),$$

$$\dot{v}_i = \boldsymbol{q}_i^T \boldsymbol{m}_i - v_i^3 - \beta_i v_i.$$

Let $p_i = \boldsymbol{q}_i^T \boldsymbol{m}_i$. Then, we have

$$\dot{p}_i = v_i \|\boldsymbol{m}_i\|^2 - v_i^2 p_i,$$

$$\dot{v}_i = p_i - v_i^3 - \beta_i v_i.$$

Finally, to get the desired form, we make the substitution $a_i = p_i / \|\boldsymbol{m}_i\|^{3/2}$ and $b_i = v_i / \|\boldsymbol{m}_i\|^{1/2}$. Hence, we get

$$\dot{a}_i = \|\boldsymbol{m}_i\| (b_i - b_i^2 a_i),$$

$$\dot{b}_i = \|\boldsymbol{m}_i\| (a_i - b_i^3 - \delta_i b_i).$$

where $\delta_i = \beta_i / \|\boldsymbol{m}_i\|$. The above equation has the same form as Equation 2. Thus, from Theorem 2 we get that, as $t \to \infty$, $(a_i(t), b_i(t))$ converges to $\pm(\sqrt{\rho_i}, \sqrt{\rho_i})$. Therefore, we get that $v_i(t)$ converges to $\pm \|\boldsymbol{m}_i\|^{1/2} \cdot \sqrt{\rho_i}$, and $p_i(t)$ converges to $\pm \|\boldsymbol{m}_i\|^{3/2} \cdot \sqrt{\rho_i}$.

Furthermore, we have

$$\ell(\boldsymbol{Q}, \boldsymbol{v}) = \frac{1}{2} \sum_{i=1}^d \|v_i \boldsymbol{q}_i - \boldsymbol{m}_i\|^2$$

$$= \frac{1}{2} \sum_{i=1}^d \left[ v_i^2 \|\boldsymbol{q}_i\|^2 + \|\boldsymbol{m}_i\|^2 - 2v_i p_i \right]$$

$$= \frac{1}{2} \sum_{i=1}^d \|\boldsymbol{m}_i\|^2 \left[ b_i^4 + \delta_i b_i^2 - 1 + 2(1 - a_i b_i) \right]$$

$$\leq \frac{1}{2} \sum_{i=1}^d \|\boldsymbol{m}_i\|^2 \left[ |\bar{\rho}_i - b_i^2| \cdot |\rho_i + b_i^2| + 2 \cdot |1 - a_i b_i| \right]$$

Thus, from Theorem 2, we get the required result. □

**Remark 1.** *We make the following remarks as extension of theorem 1.*

1. ***Bad initializations.*** *We note that the loss function decays to $0$ given that certain initial conditions hold. In particular, suppose $\boldsymbol{v}_0$ is fixed. Then the convergence to $0$ fails only if*

$$\boldsymbol{Q}_0 \in \bigcup_{i \in [d]} \{\boldsymbol{Q} : \boldsymbol{q}_i^T \boldsymbol{m}_i + \bar{\rho}_i v_i^{(0)} = 0\}.$$

   *This set is a union of hypersurfaces in the space $\mathbb{R}^{d \times d}$. Thus, with random initialization, the loss function decays to $0$ with probability $1$.*

2. ***Effect of imbalance.*** *When the conserved quantities in Equation 1 are equal to $0$ for all $i \in [d]$, we say that the inputs are "balanced". This is the condition on initialization that is assumed in Arora et al. (2019). Therefore, the numbers $\delta_i s$ can be said to measure the "imbalance" on the inputs. To isolate the contribution of these quantities to the decay of the loss function we assume that the columns of $\boldsymbol{M}$ are normalized. That is $\|\boldsymbol{m}_i\| = 1$ for all $i \in [d]$. Without loss of generality suppose $\delta_1 = \max_i \{\delta_i\}$. Then we get $\alpha = \rho_1$. Thus the decay of the loss function is controlled by the largest imbalance across the different components.*

## 5 SOLUTION TO A SPECIAL DYNAMICAL SYSTEM

We show in the previous sections that the dynamical system described in equation 2 is crucial to proving convergence of gradient descent on certain networks. This section is dedicated to explicitly finding a solution to this system. In particular, we prove the following theorem.

**Theorem 2.** *Let $x$ and $y$ be continuous functions on $(0, \infty)$ described by the dynamical system*

$$\dot{y} = c_1(x - x^2 y),$$
$$\dot{x} = c_1(y - x^3 - cx), \tag{2}$$

*where $c_1 > 0$ and $c$ are constants. Let $\rho = \sqrt{1 + \frac{c^2}{4}} + \frac{c}{2}$, $\bar{\rho} = 1/\rho$ and $2a = \rho + \bar{\rho}$. Then the limit $\lim_{t \to \infty}(x_t, y_t) = (x_\infty, y_\infty)$ exists, and it depends on the initialization $(x_0, y_0)$ as follows:*

$$(x_\infty^2, y_\infty^2) = \begin{cases} (\sqrt{\bar{\rho}}, \sqrt{\rho}) & \text{if } -\bar{\rho}x_0 \neq y_0, \\ (0, 0) & \text{otherwise.} \end{cases}$$

*Further, with $-\bar{\rho}x_0 \neq y_0$, as $x_t^2$ converges to $\bar{\rho}$ and $x_t y_t$ converges to $1$, we obtain a $T > 0$ such that for all $t > T$ the residues decay at the following rate:*

$$|\bar{\rho} - x_t^2| \text{ and } |1 - x_t y_t| = \begin{cases} \mathcal{O}(t \cdot e^{-2c_1 \cdot t}) & \text{if } c = 0, \\ \mathcal{O}(e^{-2\bar{\rho}c_1 \cdot t}) & \text{otherwise,} \end{cases}$$

*where $T$ depends on the initialization as follows:*

$$2ac_1 \cdot T = \begin{cases} -\ln \left| \frac{(\bar{\rho}x_0 + y_0)}{(\rho x_0 - y_0)} \right| & \text{if } -\bar{\rho}x_0 > y_0, \\ -\ln \frac{(\bar{\rho}x_0 + y_0)}{(\rho x_0 - y_0)} + \ln \rho^2 & \text{if } -\bar{\rho}x_0 < y_0 < 0, \\ 0 & \text{otherwise.} \end{cases}$$

*Proof sketch.* We look at the dynamical system expressed in Equations 2, and substitute the expression for $y$ from the second equation into the first equation. This gives a second-order differential equation in the function $x$. These are generally difficult to solve. However, one can discover two quantities dependent on $x$ and $y$, namely

$$r = \frac{y}{x} \text{ and } s = \frac{1 - xy}{(\rho x - y)(\bar{\rho}x + y)} = \frac{1 - xy}{x^2(\rho - r)(\bar{\rho} + r)},$$

which evolve by obeying only first order differential equations (these quantities are simplified versions of quantities which appeared while solving the second order differential equation in $x$). In particular, we get

$$\dot{r} = c_1(\rho - r)(\bar{\rho} + r) \text{ and } \dot{s} = c_1(1 + cs).$$

Upon solving these differential equations with suitable constants determined by the initial values we get

$$r_t = \rho - \frac{2a}{c_2 \cdot e^{(2ac_1)t} + 1} \text{ and } s_t = \begin{cases} \frac{1}{c}\left[c_3 \cdot e^{(cc_1)t} - 1\right] & \text{if } c \neq 0, \\ c_1 \cdot t + c_4 & \text{if } c = 0. \end{cases}$$

Now, substituting $y = rx$ in the definition of $s$ we obtain the relation

$$x^2 = [r + s(\rho - r)(\bar{\rho} + r)]^{-1}.$$

As $r$ converges to $\rho$, we get that $\bar{\rho} + r$ is eventually bounded below and above by strictly positive constants. Further, when $c \neq 0$, the function $s$ increments with an exponential rate of $cc_1$, but $\rho - r$ depletes with an exponential rate of $2ac_1$. Together, the function $s(\rho - r)$ depletes with an exponential rate of $2\bar{\rho}$. Therefore, $x^2$ converges to $\bar{\rho}$. A similar analysis is done when $c = 0$.

For the convergence of $1 - xy$ we write it as $1 - rx^2$. As we have convergence information for $r$ and $x^2$ individually, we obtain the same for $1 - xy$.

*Remark.* Given a first-order differential equation, it could be easy to find a solution that satisfies it using formal integration. However, to ensure that the quantities of interest (functions of $x$ and $y$) have the exact trajectory as the obtained solution requires uniqueness properties. This can turn out to be difficult, especially when the solution has possible points of discontinuity. A full proof of Theorem 2 appears in the appendix and addresses these issues.

## 6 ADDITIONAL RELATED WORK

Early work by Baldi & Hornik (1989) showed that for 2-layer linear networks, under very mild conditions, every local minimum is a global minimum. Furthermore, they fully characterized the saddle points in this case; in particular, all saddle points are strict (the Hessian at the saddle point has at least one strictly negative eigenvalue). Kawaguchi (2016) studied the landscape of deeper linear networks and showed that every local minimum is a global minimum for them as well; however, now saddle points need not be strict. Arora et al. (2019) gave a trajectory-based analysis gradient descent training of linear neural networks for any depth for suitable initializations. Du & Hu (2019) gave an analysis for initializations used in practice under the assumption that the network is wide: the intermediate widths of the network are much larger compared to input and output widths. Later papers Xu et al. (2023); Min et al. (2023) provided improved guarantees. But to our knowledge for depth two or more there are no guarantees without assuming either wide networks or stylized initializations.

Zhang et al. (2023) analyze 1-layer linear self-attention layer similar to the one considered in this paper. The specific problem they analyze is linear regression motivated by results in von Oswald et al. (2023). They note a connection to linear networks and require special initializations similar to Arora et al. (2018).

## 7 CONCLUSION

We analyzed the gradient flow training dynamics of linear self-attention for a natural class of histogram-like problems. Our proof depended on reduction to df-linear neural networks and an assumption on the loss function which we experimentally verified. We find our assumption to be intuitively plausible and it is a very interesting problem to prove it. Stronger versions of the assumption have the potential to lead to analysis of linear self-attention on a larger class of problems such as in-context learning for linear regression. Many other problems remain: we used parameters $\boldsymbol{W}_{QK}$ and $\boldsymbol{W}_{VF}$ instead of the original parameters. Can our techniques be combined with results on wide neural networks, such as Du & Hu (2019), to allow the original parametrization?

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

# A APPENDIX

## A.1 ADDITIONAL LEMMAS

In this section we prove two Lemmas used in the proofs of Theorem 1 and Theorem 2.

The following lemma captures how conservation laws arise from symmetries of a function. This is a well known phenomenon, e.g. see Marcotte et al. (2023); Kunin et al. (2020); Zhao et al. (2022) and related to Noether's theorem about relationship between symmetries and conserved quantities in physics.

**Lemma A.1.** *Let* $f : \mathbb{R}^{d \times d} \times \mathbb{R}^d \to \mathbb{R}$ *be a differentiable function such that it enjoys the following symmetry:*

$$f(\boldsymbol{A}\boldsymbol{C}^{-1}, \boldsymbol{C}\boldsymbol{b}) = f(\boldsymbol{A}, \boldsymbol{b})$$

*for all* $\boldsymbol{A}, \boldsymbol{b}$ *and diagonal matrices* $\boldsymbol{C}$*. Then, under gradient flow on* $f$ *at all times* $t$ *we have the following conservation:*

$$\frac{\mathrm{d}\left(\mathcal{D}(\boldsymbol{A}_t^T \boldsymbol{A}_t - \boldsymbol{b}_t \boldsymbol{b}_t^T)\right)}{\mathrm{d}t} = 0.$$

*where the function* $\mathcal{D} : \mathbb{R}^{d \times d} \to \mathbb{R}^{d \times d}$ *retrieves the diagonal part of the matrix i.e.* $\mathcal{D}(\boldsymbol{A}) = \mathrm{Diag}(\mathrm{diag}(\boldsymbol{A}))$.

*Proof.* Fix any $t$. Let $\boldsymbol{S}$ be any arbitrary matrix. Consider the function $g(x)$ depending on the diagonal matrix $\boldsymbol{S}$, defined in a neighborhood of 0 as

$$g(x) = f(\boldsymbol{A}(x), \boldsymbol{b}(x)) = f(\boldsymbol{A}_t e^{-x\boldsymbol{S}}, e^{x\boldsymbol{S}} \boldsymbol{b}_t) = f(\boldsymbol{A}_t, \boldsymbol{b}_t).$$

Differentiating both sides with respect to $x$ at the point 0 we get

$$
\begin{aligned}
0 = g'(0) &= \left\langle \tfrac{\mathrm{d}f}{\mathrm{d}\boldsymbol{A}}\left(\boldsymbol{A}(0), \boldsymbol{b}(0)\right), \tfrac{\mathrm{d}\boldsymbol{A}}{\mathrm{d}x}(0) \right\rangle + \left\langle \tfrac{\mathrm{d}f}{\mathrm{d}\boldsymbol{b}}\left(\boldsymbol{A}(0), \boldsymbol{b}(0)\right), \tfrac{\mathrm{d}\boldsymbol{b}}{\mathrm{d}x}(0) \right\rangle \\
&= \left\langle \tfrac{\mathrm{d}f}{\mathrm{d}\boldsymbol{A}}\left(\boldsymbol{A}_t, \boldsymbol{b}_t\right), -\boldsymbol{A}_t \boldsymbol{S} \right\rangle + \left\langle \tfrac{\mathrm{d}f}{\mathrm{d}\boldsymbol{b}}\left(\boldsymbol{A}_t, \boldsymbol{b}_t\right), \boldsymbol{S}\boldsymbol{b}_t \right\rangle \\
&= \left\langle \tfrac{\mathrm{d}\boldsymbol{A}_t}{\mathrm{d}t}, \boldsymbol{A}_t \boldsymbol{S} \right\rangle - \left\langle \tfrac{\mathrm{d}\boldsymbol{b}_t}{\mathrm{d}t}, \boldsymbol{S}\boldsymbol{b}_t \right\rangle \\
&= \left\langle \boldsymbol{A}_t^T \tfrac{\mathrm{d}\boldsymbol{A}_t}{\mathrm{d}t} - \tfrac{\mathrm{d}\boldsymbol{b}_t}{\mathrm{d}t} \boldsymbol{b}_t^T, \boldsymbol{S} \right\rangle.
\end{aligned}
$$

As the above holds for every diagonal matrix $\boldsymbol{S}$, we conclude that $\mathcal{D}\left(\boldsymbol{A}_t^T \tfrac{\mathrm{d}\boldsymbol{A}_t}{\mathrm{d}t} - \tfrac{\mathrm{d}\boldsymbol{b}_t}{\mathrm{d}t} \boldsymbol{b}_t^T\right) = 0$. Finally, we observe that

$$\frac{\mathrm{d}\mathcal{D}\left(\boldsymbol{A}_t^T \boldsymbol{A}_t - \boldsymbol{b}_t \boldsymbol{b}_t^T\right)}{\mathrm{d}t} = \mathcal{D}\left(\boldsymbol{A}_t^T \tfrac{\mathrm{d}\boldsymbol{A}_t}{\mathrm{d}t} - \tfrac{\mathrm{d}\boldsymbol{b}_t}{\mathrm{d}t} \boldsymbol{b}_t^T\right) + \mathcal{D}\left(\boldsymbol{A}_t^T \tfrac{\mathrm{d}\boldsymbol{A}_t}{\mathrm{d}t} - \tfrac{\mathrm{d}\boldsymbol{b}_t}{\mathrm{d}t} \boldsymbol{b}_t^T\right)^T = 0.$$

$\square$

For the proof of Theorem 2, we need the following lemma.

**Lemma A.2.** *Suppose* $f$ *and* $h$ *are continuous functions defined in the interval* $\mathcal{I}$ *where* $\dot{f}_t$ *satisfies* $\dot{f}_t = h_t \cdot f_t$*. Then,* $f_a = 0$ *for some* $a \in \mathcal{I} \implies f_t = 0$ *on* $\mathcal{I}$*.*

*Proof.* Let $H$ be the anti-derivative of $h$ on $\mathcal{I}$. Then observe that at all $t \in \mathcal{I}$ we have

$$\frac{\mathrm{d}f e^{-H}}{\mathrm{d}t} = -fh e^{-H} + fh e^{-H} = 0.$$

Hence $f = c \cdot e^H$ for some constant $c$. Plugging the condition $f_a = 0$ gives $c = 0 \implies f \equiv 0$. $\square$

## A.2 PROOF OF THEOREM 2

In the proof sketch of Theorem 2 we have defined the quantities $r$ and $s$ as functions of $x$ and $y$, and obtained their solutions via formal integration of partial fractions. However, as mentioned earlier, this cannot be done rigorously as the integrands which appear in these equations may not be defined at several points. So, here we take a reverse approach. We start by defining $r$ and $s$ as the solutions obtained through formal integration. We go on to show rigorously that the desired functions of $x$ and $y$ must align with the functions $r$ and $s$.

*Proof. (Theorem 2)* Let us first consider $x_0, y_0 \neq 0$. We would like to keep track of the trajectory of the ratio $\frac{y_t}{x_t}$. First consider the time varying function $r$ defined according to its initial value $r_0$ as follows

$$r_t = \begin{cases} \rho - \frac{2a}{c_2 \cdot e^{(2ac_1)t} + 1} & \text{if } r_0 \neq \rho, \\ \rho & \text{if } r_0 = \rho, \end{cases} \tag{3}$$

where $c_2 = \frac{2a}{(\rho - r_0)} - 1$ when $r_0 \neq \rho$. Then $r_t$ has the following derivative:

$$\dot{r} = c_1(\rho - r)(\bar{\rho} + r).$$

When $r_0 \geq -\bar{\rho}$, equivalently $c_2 \notin [-1, 0)$, the function $r_t$ is continuous everywhere. So $(rx - y)$ is also continuous everywhere. Setting $r_0 = \frac{y_0}{x_0}$ and computing the derivative of $(rx - y)$ gives us

$$\tfrac{d(rx-y)}{dt} = -c_1(r + x^2)(rx - y) \text{ and } r_0 \cdot x_0 - y_0 = 0.$$

Thus $(rx - y)$ satisfies the requirements of Lemma A.2 giving $r_t x_t = y_t$ for all $t$.

When $r_0 < -\bar{\rho}$, equivalently $c_2 \in (-1, 0)$, then $r$ is discontinuous at the point $T = \frac{-(\ln |c_2|)}{2ac_1}$. But this time $r^{-1}$ is continuous everywhere. Thus applying Lemma A.2 in a similar fashion to $(x - r^{-1}y)$ gives us $x_t = r_t^{-1} y_t$ for all $t$. One can see that $r_t^{-1} = 0$ only at the point $T$. Therefore, dividing by $r_t^{-1}$ gives $r_t x_t = y_t$ for all $t \neq T$. Therefore we get a complete characterization of the relation between $x_t, y_t$ and $r_t$ as follows:

$$r_t \cdot x_t = \begin{cases} y_t & \text{for all } t \text{ if } \frac{y_0}{x_0} \geq -\bar{\rho}, \\ y_t & \text{when } t \neq \frac{-(\ln |c_2|)}{2ac_1} \text{ if } \frac{y_0}{x_0} < -\bar{\rho}. \end{cases} \tag{4}$$

Note that we have used $x_0, y_0 \neq 0$ to initialize $r_0$ as required above. In particular, we have used $x_0 \neq 0$ to define $r_0 \geq -\bar{\rho}$ and used $y_0 \neq 0$ to define $r_0 < -\bar{\rho}$. Let us now consider the following cases.

**Case 1.** Suppose $\frac{y_0}{x_0} \notin \{\rho, -\bar{\rho}\}$. Here consider the time varying functions $s$ given by

$$s_t = \begin{cases} \frac{1}{c}\left[c_3 \cdot e^{(cc_1)t} - 1\right] & \text{if } c \neq 0 \\ c_1 \cdot t + c_4 & \text{if } c = 0, \end{cases}$$

where the constants $c_3$ and $c_4$ can be fixed by fixing the initial values $s_0$ as follows: $c_3 = 1 + c \cdot s_0$ and $c_4 = s_0$. We further note that the derivative of $s$ is given by

$$\dot{s} = c_1(1 + cs).$$

First note that $s$ is continuous everywhere on $(0, \infty)$. Then the map $g := s(\bar{\rho}x + y)(\rho x - y) - (1 - xy)$ is also continuous everywhere. We fix $s_0$ as given below and also compute the derivative of $g$ as follows

$$s_0 = \frac{1 - x_0 \cdot y_0}{x_0^2(\rho - \frac{y_0}{x_0})(\bar{\rho} + \frac{y_0}{x_0})} \text{ and } \dot{g} = -c_1 x^2 g.$$

We have used the conditions $\frac{y_0}{x_0} \notin \{\rho, -\bar{\rho}\}$ of this case in fixing $s_0$ as above. Thus, the requirements for applying Lemma A.2 are satisfied, and as a result we have $g \equiv 0$. That is,

$$s_t(\rho x_t - y_t)(\bar{\rho}x_t + y_t) = 1 - x_t y_t \text{ for all } t. \tag{5}$$

For sufficiently large $t$ the equation $r_t \cdot x_t = y_t$ is defined (as stated in equation 4). So we substitute $y_t = r_t \cdot x_t$ in the expression for $s_t$ in the above equation 5 to get

$$1 - r_t x_t^2 = s_t(\rho - r_t)(\bar{\rho} + r_t)x_t^2$$
$$\implies x_t^2 = \left[r_t + s_t(\rho - r_t)(\bar{\rho} + r_t)\right]^{-1}. \tag{6}$$

*Convergence Analysis.* With these expressions in place, we are all set to investigate the limiting behavior of $x_t$ and $y_t$. Let $x_\infty, y_\infty$ and $r_\infty$ be the limits of $x_t, y_t$ and $r_t$ respectively. We will show that these limits exist. We will see that we need $r_t$ to be bounded away from $-\bar{\rho}$ and 0. For this we simply look at the expression for $r_t$ under different initializations, and find a $T$ in every case such that for all $t > T$ this required property of being bounded away from 0 and $-\bar{\rho}$ holds. We note that the monotonicity of $r_t$ helps us make this argument. We organize the observation in the following table.

| $r_0 = \frac{y_0}{x_0}$ | $c_2 = (\bar\rho + r_0)/(\rho - r_0)$ | $T$ | $r_t$ on $(T, \infty)$ | $r_t$ |
|---|---|---|---|---|
| $(-\infty, -\bar\rho)$ | $(-1, 0)$ | $-(\ln|c_2|)/2ac_1$ | decreasing | $(\rho, r_T)$ |
| $(-\bar\rho, 0]$ | $(0, \bar\rho^2]$ | $(\ln\bar\rho^2 - \ln c_2)/2ac_1$ | increasing | $(r_T, \rho)$ |
| $(0, \rho)$ | $(\bar\rho^2, \infty)$ | $0$ | increasing | $(r_0, \rho)$ |
| $(\rho, \infty)$ | $(-\infty, -1)$ | $0$ | decreasing | $(\rho, r_0)$ |

Note that $r_t$ being bounded away from $-\bar\rho$ means that $\left|\frac{r_t}{(\bar\rho + r_t)}\right|$ is bounded, for sufficiently large $t$ depending on the initialization as stated in the above table. We use this and the expression for $x^2$ in equation 6 to observe that

$$|1 - xy| = |1 - rx^2| = \left|1 + \frac{r/(\bar\rho + r)}{s(\rho - r)}\right|^{-1} = \mathcal{O}\left(|s(\rho - r)|\right).$$

Similarly, we further use $r_t$ being bounded away from $0$ for the following residue

$$
\begin{aligned}
|\bar\rho - x^2| &= \bar\rho\left|1 - rx^2 + \tfrac{r-\rho}{r}(rx^2 - 1 + 1)\right| \\
&\leq \bar\rho|1 - rx^2| + \bar\rho\left|\tfrac{r-\rho}{r}\right|\left(1 + |1 - rx^2|\right) \\
&= \mathcal{O}\left(|1 - rx^2|\right) \\
&= \mathcal{O}\left(|s(\rho - r)|\right).
\end{aligned}
$$

Thus, $s(\rho - r)$ is the quantity of interest, and therefore we study its limiting behavior. When $c \neq 0$, we have

$$|s_t(\rho - r_t)| = \frac{2a}{|c|}\left|\frac{c_3 \cdot e^{c(c_1 t)} - 1}{c_2 \cdot e^{2a(c_1 t)} + 1}\right| \leq c_5 \cdot e^{-(2\bar\rho c_1)t},$$

where $c_5 > 0$ is some constant. Here we have used $2a - c = 2\bar\rho$. Similarly, when $c = 0$, we do a similar calculation to obtain

$$|s_t(\rho - r_t)| = 2a\left|\frac{c_1 t + c_4}{c_2 \cdot e^{2c_1 t} + 1}\right| \leq c_6 \cdot t \cdot e^{-(2c_1)t},$$

for some constant $c_6 > 0$. In either case, we therefore conclude that $\lim_{t\to\infty} s_t(\rho - r_t) = 0$. As $|s(\rho - r)|$ decays to $0$ we get that $x_\infty = \pm\sqrt{\rho}$ and $x_\infty \cdot y_\infty = 1$. Thus, $y_\infty = \pm\sqrt{\rho}$ with the same sign as $x_\infty$ (as $r_\infty$ positive). This case is hence complete.

**Case 2.** Suppose $\frac{y_0}{x_0} = \rho$. Then equation 3 tells us that $r = \rho$ for all $t$. Substituting $y = \rho x$ in $\dot{x}$ gives the equation

$$\dot{x} = c_1 x(\bar\rho - x^2) \implies \frac{dx^2}{dt} = 2c_1 x^2(\bar\rho - x^2).$$

Here consider the time varying function $X$ defined as

$$X_t = \frac{\bar\rho}{1 + c_2 e^{-2c_1\bar\rho t}}$$

where $c_2 = \frac{\bar\rho}{X_0} - 1$. Setting $X_0 = x_0^2$ we get that $X_t$ is continuous everywhere as $c_2 > -1$. For the function $(X_t - x_t^2)$ we have an initial condition equalling $0$ and the derivative condition

$$\frac{d(X - x^2)}{dt} = 2c_1(x^2 + X + \bar\rho)(X - x^2).$$

Thus we apply Lemma A.2 obtaining $x_t^2 = X_t$ at all $t$. We also have that $\lim x^2 = \bar\rho$ where $\left|\bar\rho - x^2\right|$ decays with an exponential rate of $-2\bar\rho c_1$. This is the same limit and convergence rate which we obtain in Case 1.

**Case 3.** Suppose $\frac{y_0}{x_0} = -\bar\rho$. Equation 3 tells us that $r_t = -\bar\rho$ for all $t$. This reduces $\dot{x}$ to $\dot{x} = -(\rho + x^2)x \leq -\rho \cdot x$, thus ensuring that $x$ decays to $0$ with an exponential rate of at least $\rho$. Thus, in this case $(x_\infty, y_\infty) = (0, 0)$.

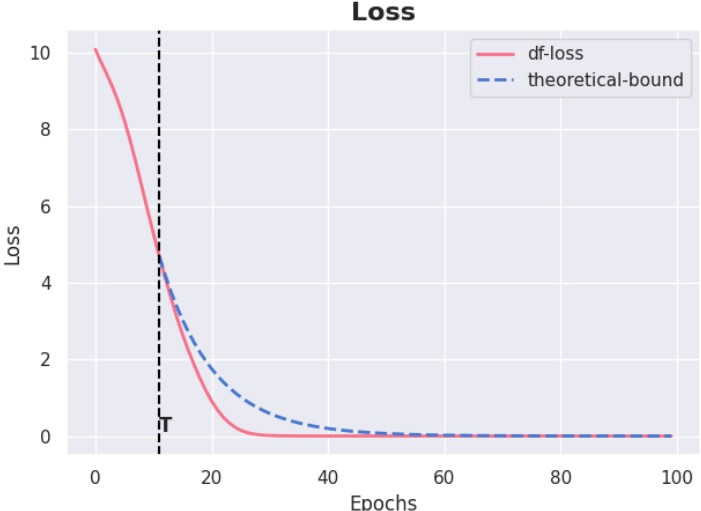

Figure 3: Training loss $\ell_{\mathrm{df}}$ and the theoretical bound for the loss given in Theorem 1, for `Equality Histogram` trained using gradient descent starting from random initialization. We see the exponential convergence of the loss to 0, as described in Theorem 1.

The only initial case that remains is when $x_0 = 0$ or $y_0 = 0$. Suppose $x_0 = 0$ and $y_0 \neq 0$. Then note that $\dot{x}_0 = c_1 y_0 \neq 0$. Then for very small $\epsilon > 0$ we get $x_\epsilon, y_\epsilon \neq 0$. Thus we can apply the same convergence analysis with our initial assumption of nonzero $x$ and $y$ starting from time $\epsilon$. A similar analysis can be done for $x_0 \neq 0$ and $y_0 = 0$.

Finally if $x_0 = y_0 = 0$, the function doesn't change. The collation of these steps complete the proof for theorem 2. □

### A.3 ADDITIONAL EXPERIMENTS

In this section we describe some additional experiments to validate the claims presented.

**Random Initialization** To experimentally validate the theoretical results presented, we train a DF-Linear network to minimize the $\ell_{\mathrm{df}}$ loss on the `EqualityHistogram` task, using gradient descent. Figure 3 shows the training loss and the theoretical bounds derived in Theorem 1 starting from random Gaussian initialization. Note that, since gradient descent is a discretization of gradient flow, the time step $N$ post which we expect to see exponential decay was calculated as $N = T/\eta$.

**Initializing Close to the Bad Initializations** From Remark 1, we know the set of bad initializations - initializations that lead us to saddle points. For example, for the `Equality Histogram` task, one such set of points is $\boldsymbol{v}_0 = c\mathbf{1}_d$ and $\boldsymbol{Q}_0 = -c\boldsymbol{I}_d$, for $c \in \mathbb{R}$ i.e. the set of points such that $\boldsymbol{q}_i^{(0)^T} \boldsymbol{m}_i + \bar{\rho}_i v_i^{(0)} = 0$ for all $i \in [d]$. Initializing from this set, the trajectory converges to the saddle point at $\boldsymbol{Q} = 0, \boldsymbol{v} = 0$.

Figure 4 shows the trajectories of two systems, one initialized close to this set (satisfying $\boldsymbol{q}_i^{(0)^T} \boldsymbol{m}_i + \bar{\rho}_i v_i^{(0)} = 0.01$), and the other initialized slightly farther away from this set (satisfying $\boldsymbol{q}_i^{(0)^T} \boldsymbol{m}_i + \bar{\rho}_i v_i^{(0)} = 0.1$), with all other parameters same. We see that, for initializations close to the set of points that lead to the saddle point, it takes longer to converge - we remain close to the saddle point for longer. This can also be seen theoretically - the time $T$ following which exponential decay occurs is much larger for initializations close to the set of bad initializations, and similarly, the rate of decay is much smaller.

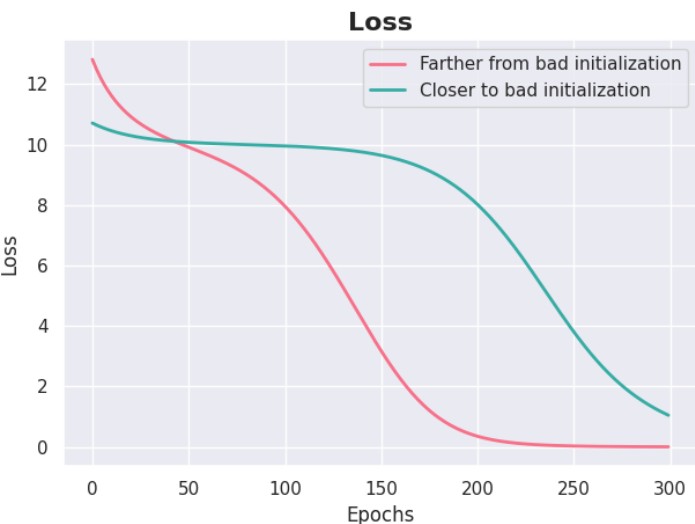

Figure 4: Loss curves for gradient descent on $\ell_{\mathrm{df}}$ starting from different distances to the set of points that lead to a saddle point. Note that initializations closer to the set of bad initializations get closer to the saddle point, and take longer to escape.

