# OpenReview forum: "Towards Analyzing Self-attention via Linear Neural Network"
_ICLR.cc/2024/Conference — Submitted to ICLR 2024_

### Official Review · Reviewer_7ZKc · 2023-11-01

**Soundness:** 3 good
**Presentation:** 3 good
**Contribution:** 2 fair
**Rating:** 5
**Confidence:** 4

**Summary:**

The paper analyzes the gradient flow training dynamics of a simplified linear transformer on the histogram task. The method reduces the training of a simplified transformer to that of a linear neural network with two layers where the first layer is a diagonal matrix. The theoretical results of the paper are based on one assumption, which is experimentally justified.

**Strengths:**

1. The proofs of the theorems and lemmas are provided and detailed.

2. Assumption 3.1 for the theoretical results seems reasonable and experimentally justified.

3. The paper is well-motivated.

**Weaknesses:**

1. The paper only deals with a very simple attention layer with only a single linear layer and lacks the components of the transformer model. This setting is not practical in real-world applications and thus limits the scope of the paper’s results.

2. The paper only considers the histogram tasks, which is rather limited in the context of Transformers.

3. The paper lacks experiment results to demonstrate the theoretical results.

**Questions:**

1. Can the results of the paper be extended to other common machine learning tasks where Transformers succeed such as language modeling or machine translation, rather than just the histogram tasks?

2. It would be helpful to show the decay behavior of the loss function $l$ in Theorem 1 under random initialization. Additionally, the authors should demonstrate the behavior of $l$ under bad initialization.

---

> ### Author Response · Authors · 2023-11-20
>
> Thank you for your suggestions. We have added additional some additional experiments demonstrating our theoretical results, and the behavior of the loss function under random and bad initializations.

---

### Official Review · Reviewer_nN7u · 2023-11-06

**Soundness:** 2 fair
**Presentation:** 2 fair
**Contribution:** 2 fair
**Rating:** 3
**Confidence:** 3

**Summary:**

This paper aims to understand the training dynamics of self-attention networks. In particular, the paper focuses on a simplified single-layer self-attention network without softmax, MLP, layer normalization, and positional embeddings. It restricts itself to a specific class of learning tasks, namely histogram-like tasks. In its simplest form, given an $N$ length sequence, this task requires the network to produce an $N$ length output sequence where $i$th output element contains the frequency of the input element at the $i$th position in the input sequence. The paper reduces the problem of learning the simplified self-attention model to learning a two-layer linear network. Subsequently, the paper analyzes the gradient flow for learning the two-layer linear network.

**Strengths:**

1) The paper considers multiple variants of the histogram-like learning tasks for a single-layer self-attention model.
2) The paper explores a connection between learning self-attention models and linear networks.
3) The paper exploits the structure of the underlying problem and breaks down the gradient flow analysis of learning linear networks to multiple one-dimensional problems.

**Weaknesses:**

1) The main setup studied in the paper is not well motivated. Why should one care about the histogram-like learning tasks? By construction, self-attention can easily model this task. But that alone does not justify the importance of such tasks.
2) The paper considers a very simplified setup, e.g., single-layer, no positional embeddings, and the size of the alphabet equal to the embedding dimension. How the findings of this paper affect the practice is not at all clear from the current version of the paper.
3) While discussing the prior works, the paper states "...While insightful, these papers generally involve stylized assumptions and this makes it difficult to compare the results." However, this paper goes on to study a completely new problem (again with various assumptions); hence does not provide any comparison with prior art.
4) There is significant scope for improvement in the presentation of the paper. For example, one can improve the flow of the paper by better organizing the key contributions and the discussion of prior work. Similarly, there is room for improvement in the presentation of the technical content. Section 4 repeatedly mentions Eq. (2) which is only introduced later in Section 5. Similarly, Theorem 2 is mentioned multiple times before being formally introduced or informally discussed. How do various points in Remark 1 constitute an as "extensions of theorem 2"?

**Questions:**

See the comments under the weaknesses section.

---

> ### Author Response · Authors · 2023-11-17
>
> Thank you for your suggestions. The remarks are better called corollaries than extensions. We will incorporate your suggestions in our revision.

---

> > ### Comment · Reviewer_nN7u · 2023-11-22
> > **Thank you for your response.**
> >
> > Thank you for your response. I have decided to keep my original score. I would encourage the authors to submit a revised version to a future venue.

---

### Official Review · Reviewer_4ez9 · 2023-11-06

**Soundness:** 2 fair
**Presentation:** 2 fair
**Contribution:** 1 poor
**Rating:** 5
**Confidence:** 4

**Summary:**

This paper simplifies the training dynamics problem of a 1-layer linear self-attention layer into the joint optimization problem with two matrix variables that minimize loss like l_{df}(Q, v) = 0.5 * |Q*Diag(v) - M|_F^2. They show that this loss will decrease in exponential speed. They also try histogram tasks and show that the learned attention maps match their expectation.

**Strengths:**

1. The writing is clear, and the main result is easy to understand.
2. I think the construction of s for solving dynamics systems in theorem 2 is skillful.

**Weaknesses:**

1. I think the total contribution of this work is not enough to be accepted by ICLR. Although this paper solves a particular dynamical system in detail with some clever construction of $r$ and $s$, which I think is the only novel contribution of this paper, the problem setting is too simple (linear attention without softmax layer, l2 loss, adding assumption 3.1 to connect l_{tf} and l_{df}, histogram tasks experiment, etc.) to give much insight to understand the true mechanism of the transformer. Several recent works(for example, [1,2,3]) mentioned in the related work part have shown that the training dynamics of the 1-layer transformer with softmax layer will let the attention map show some sparsity pattern and focus on particular topics of the input data. And their results may have covered more insightful results

[1] Yuchen Li, Yuanzhi Li, and Andrej Risteski. How do transformers learn topic structure: Towards a mechanistic understanding, 2023.
[2] Yuandong Tian, Yiping Wang, Beidi Chen, and Simon Du. Scan and snap: Understanding training dynamics and token composition in 1-layer transformer, 2023.
[3] Samet Oymak, Ankit Singh Rawat, Mahdi Soltanolkotabi, and Christos Thrampoulidis. On
the role of attention in prompt-tuning. arXiv preprint arXiv:2306.03435, 2023.

**Questions:**

The same as weakness

---

### Author Response · Authors · 2023-11-17
**Common Response to the Reviewers**

We thank the reviewers for their responses.

A common concern of the reviewers is that the problem considered is too simple because (1) histogram-like tasks are very simple, (2) the self-attention model strips many features of transformers like softmax.

Theoretical analysis of deep learning lags far behind empirical model. Even one-hidden-layer MLPs are not well-understood. This has necessitated that theoretical analyses simplify the problem to its core, e.g., by considering toy learning problems and simplifying the transformer, or by introducing multiple assumptions.

For example, recently the problem of analysing linear regression using one-layer transformers has attracted much attention, e.g. [4, 5, 6], and these papers study simplified models similar to ours.

1. Similarly, in the papers suggested by Reviewer 4ez9, either the results are not as strong as ours, or the proofs require stronger assumptions than ours:
	1. In [1], there are no guarantees on convergence or convergence rates. Furthermore, all the results assume that some of the parameters are fixed during training. In contrast, we provide results under no such assumptions.
	2. In [2], the results rely on the assumptions that the input sequence is long enough, and that the learning rates for decoder layer and attention layer are different, and one is much faster than the other.
	3. In [3], there are multiple assumptions and it studies a related but distinct problem, namely prompt tuning.

While these results are incomparable and so is ours, we think the problem of understanding transformer training dynamics, and more generally of neural networks, is a difficult and multifaceted one, and these papers are attacking different facets. Thus not being comparable remark in our paper was not meant as a  criticism of prior work.



Linear neural networks
Apart from the analysis of self-attention, our work also contributes to the analysis of linear neural networks as discussed in our paper. Perhaps surprisingly, even the analysis of two-layer linear neural networks is currently incomplete despite considerable work in the last few year. Further simplified models such as diagonal linear networks are studied in the literature, e.g., [Vaskevicius et al. (2019); Woodworth et al. (2020); HaoChen et al. (2021); Pesme et al. (2021); Berthier (2023); Boix-Adsera et al. (2023)] as cited in our paper.

Furthermore, recent work suggests that analyzing linear transformers (without softmax) is helpful even for studying more general transformers [5].


[1] Yuchen Li, Yuanzhi Li, and Andrej Risteski. How do transformers learn topic structure: Towards a mechanistic understanding, 2023.

[2] Yuandong Tian, Yiping Wang, Beidi Chen, and Simon Du. Scan and snap: Understanding training dynamics and token composition in 1-layer transformer, 2023.

[3] Samet Oymak, Ankit Singh Rawat, Mahdi Soltanolkotabi, and Christos Thrampoulidis. On the role of attention in prompt-tuning. arXiv preprint arXiv:2306.03435, 2023.

[4] J. von Oswald et al. Transformer learn in-context by gradient descent. ICML 2023.

[5] Ahn, K., Cheng, X., Daneshmand, H. and Sra, S., 2023. Transformers learn to implement preconditioned gradient descent for in-context learning. NeurIPS 2023.

[6] R. Zhang et al. Trained Transformers Learn Linear Models In-Context. https://arxiv.org/abs/2306.09927

[7] Ahn, K., Cheng, X., Song, M., Yun, C., Jadbabaie, A. and Sra, S., 2023. Linear attention is (maybe) all you need (to understand transformer optimization). arXiv preprint arXiv:2310.01082.

---

### Meta-Review · Area_Chair_oRpT · 2023-12-01

**Metareview:**

All reviewers find the paper making too many simplifications to Attention architecture as well as the training task for the presented analysis to be of interest. Given the existing work on analyzing transformer dynamics, I agree with reviewers that the setting of this work is not very interesting and hence recommend rejection.

**Justification For Why Not Higher Score:**

Not an interesting setting for analyzing Transformers.

**Justification For Why Not Lower Score:**

N/A

---

### Decision · Program_Chairs · 2024-01-16

Reject